# Renormalization-group improvement in hadronic $\tau$ decays in 2018

**Diogo Boito[1*], Pere Masjuan[2] and Fabio Oliani[1]**

**1** Instituto de Física de São Carlos, Universidade de São Paulo,
CP 369, 13560-970, São Carlos, SP, Brazil
**2** Grup de Física Teòrica, Departament de Física, Universitat Autònoma de Barcelona,
and Institut de Física d'Altes Energies (IFAE), The Barcelona Institute of Science
and Technology (BIST), Campus UAB, E-08193 Bellaterra (Barcelona), Spain

⋆ boito@ifsc.usp.br

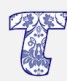 *Proceedings for the 15th International Workshop on Tau Lepton Physics,
Amsterdam, The Netherlands, 24-28 September 2018*

## Abstract

One of the main sources of theoretical uncertainty in the extraction of the strong coupling from hadronic tau decays stems from the renormalization group improvement of the series. Perturbative series in QCD are divergent but are (most likely) asymptotic expansions. One needs knowledge about higher orders to be able to choose the optimal renormalization-scale setting procedure. Here, we discuss the use of Padé approximants as a model-independent and robust method to extract information about the higher-order terms. We show that in hadronic $\tau$ decays the fixed-order expansion, known as fixed-order perturbation theory (FOPT), is the most reliable mainstream method to set the scale. This fully corroborates previous conclusions based on the available knowledge about the leading renormalon singularities of the perturbative series.

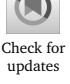

# 1 Introduction

Since the 1990s, inclusive hadronic decays of the $\tau$ lepton have been acknowledged as a reliable source of information about QCD. In particular, the strong coupling, $\alpha_s$, can be extracted with competitive precision from these decays. Since the works by Braaten, Narison and Pich [1] and later by Le Diberder and Pich [2], which finally shaped the standard strategy to extract $\alpha_s$ from this process, several important developments have occured. On the experimental side, the precision has improved a lot thanks to the LEP experiments; the latest (re)analysis of ALEPH data was published in 2014 [3]. On the theory side, our understanding of the theoretical input from QCD necessary to achieve an accurate $\alpha_s$ determination has improved as well. In parallel, there was similar progress in the global knowledge about $\alpha_s$ from other processes in the past 25 years. The uncertainty in the PDG recommendation for $\alpha_s(m_Z)$ went down from about 5% in 1994 [4] to a mere 0.9% in the latest edition [5,6], while individual extractions from the lattice are achieving uncertainties below 1% (see, for example, [7]). Although the extraction of $\alpha_s$ from $\tau$ decays remains appealing — it is performed at rather low-energies and provides, therefore, a non-trivial test of asymptotic freedom — it must be carefully scrutinized given the state of affairs.

In the last few years, a reassessment of the $\alpha_s$ extraction from $\tau$ decays was motivated by the publication of the result for the $\alpha_s^4$ correction in the relevant perturbative QCD series, which is the next-to-next-to-next-to-leading order (N3LO) correction [8,9]. This tour de force calculation, a five-loop QCD result involving about 20,000 Feynman diagrams, was completed in 2008 — more than 15 years after the publication of the $\alpha_s^3$ result [10,11]. Since then, many aspects of the extraction of $\alpha_s$ from $\tau$ decays have been reexamined. In this note, we will focus on the perturbative series, in particular on its renormalization group improvement.

The QCD description of hadronic tau decays must rely on finite-energy sum rules, which exploit analyticity in order to circumvent the breakdown of perturbative QCD at low energies. The theoretical predictions are then obtained from a contour integral in the complex plane of the variable $s$ — the invariant mass of the final-state hadrons. When performing this integration, one must set the renormalization scale. The two most common procedures are known as fixed-order perturbation theory (FOPT), in which the scale is kept fixed, and contour-improved perturbation theory (CIPT) [2,12], in which the scale runs along the contour of integration. The two lead to different series and this difference, which is larger than the error ascribed to each series individually, is one of the main sources of theoretical uncertainty in the extraction of $\alpha_s$ from hadronic $\tau$ decay data.

Before entering the specifics of $\tau$ decays let us remind some basic facts about perturbative expansions in QCD. As discovered by Dyson in 1952, the perturbative series in powers of the coupling in realistic quantum field theories are divergent expansions [13], no matter how small the coupling is. The fact that the first few terms of these series do provide meaningful results, i.e. they seem to agree reasonably well with experiment, led Dyson to conjecture that these series must be asymptotic expansions: a special type of divergent series that are useful in practice. Asymptotic expansions approach the true value of the function being expanded up to a finite order, after which the series starts to diverge. Their usefulness is illustrated by the famous Carrier's rule which states that

> "Divergent series converge faster than convergent series because they don't have to converge" [14].

The idea is that the series may approach the true value much faster than a convergent expansion, which is actually a fortunate feature in QCD, since the computation of higher-order corrections becomes quickly impractical. It also implies that a good asymptotic expansion, in comparison with a convergent one, can have consecutive terms that decrease less in magnitude when compared to their predecessor, precisely because "it does not have to converge".

The divergence of the series is due to the factorial growth of its coefficients — this behaviour, in turn, can be mapped to regions of specific loop diagrams. The "optimal truncation" of an asymptotic series of this type is often achieved by truncating it at its smallest term. This goes under the name superasymptotic approximation [14]. The error that is made in such an approximation is typically of the order of $e^{-p/\alpha}$, where $p > 0$ is a constant and $\alpha$ the expansion parameter. The quantity $e^{-p/\alpha}$ is non-perturbative and does not admit a power series in $\alpha$, but vanishes when the expansion parameter goes to zero, as one would intuitively expect.[1] This suggests that the issues related to the fact that the series is asymptotic become more prominent when the coupling is larger. In QCD this means lower energies, such as in $\tau$ decays where the relevant scale, of the order of the $\tau$ mass is $\sim 2$ GeV. One should say that these rules do not have the status of theorems, mathematical proofs are rare here. As a matter of fact, there is no proof that the series in QCD is asymptotic to start with, but everything indicates that this is indeed the case.

The discussion about FOPT and CIPT in $\tau$ decays and a decision about which one is the most reliable procedure cannot be taken out of this context. They are both asymptotic series (at best), therefore divergent. In particular, some of the arguments put forward in the literature in favour of CIPT [18][2] based on the relative size of the first few terms of the series are insufficient, since they tend to downplay, or simply ignore, these basic features of perturbative expansions in QCD. Also, in analysing the series it does not make sense to talk about a radius of convergence, because we are dealing with divergent expansions. Inevitably, a final conclusion about the reliability of the two different procedures requires knowledge about higher-order coefficients of the series. In the absence of higher-order loop computations, one must resort to other methods to estimate those terms.

A possibility is to use our partial knowledge about the renormalons of the series. It is well known that the behaviour of a series of this type at intermediate and high orders is dominated by the renormalons close to the origin. Under reasonable assumptions, one can then construct an approximation to the Borel transform of the series[3] using the leading renormalons and match this description to the exactly know coefficients. This allows for an extrapolation to higher orders and one is able to obtain an estimate for the higher-order coefficients. These type of construction has been studied in detail in Refs. [19, 20]. The main conclusions that can be drawn from this strategy are twofold.

1. Under reasonable assumptions, i.e., without any artificial suppression of leading renormalon singularities, FOPT is the most reliable method to set the renormalization scale in hadronic $\tau$ decays. Because in CIPT a subset of terms, associated with the running of the coupling, are resummed to all orders important cancellations are missed and the series does not provide a good approximation to the "true" value — understood as the value obtained from the Borel sum of the reconstructed series.

2. The fact that FOPT is to be preferred is linked to the renormalon singularity associated with the gluon condensate. Should this singularity be, for some unknown reason, much suppressed then CIPT would be best.

These conclusions, albeit providing strong support to FOPT, are somewhat model dependent since they do rely on the partial knowledge about the renormalons and could be affected by

---

[1]Given the logarithmic running of $\alpha_s$ in QCD the error of the truncated perturbative QCD expansion becomes $e^{-p/\alpha_s(Q)} \sim \left(\frac{\Lambda^2}{Q^2}\right)^p$. These non-perturbative power corrections in $1/Q^2$ are, of course, related to the higher-order terms in the Wilson's OPE. In the case of $\tau$ decays, the leading one is given by the gluon condensate and scales as $1/Q^4$. There is an infinite series of such terms, one for each gauge-invariant operator that contributes to the OPE and, therefore, the QCD expansion becomes a double expansion in $\alpha_s$ and in $1/Q^2$ [15]. Effects related to asymptotic nature of the latter are related to the so-called duality violations [16,17].

[2]This argument [18] is essentially unaltered since the publication of Ref. [2] in 1992.

[3]Essentially its inverse Laplace transform.

the inclusion or the removal of a specific singularity from the model. It is, therefore, desirable to study this issue from a model-independent point of view in order to corroborate, or to discredit, the results obtained from the renormalon models.

Here we will discuss recent results presented in Ref. [21] where we used the mathematical method of Padé approximants [22–24] to extract information about the higher-order coefficients of the series. Padé, or rational, approximants are a reliable model-independent tool that has regained importance in recent years and has found applications in many aspects of particle physics [25–29]. In Ref. [21] we applied the method systematically to the problem of estimating higher orders in the perturbative QCD description of hadronic $\tau$ decays. We first used the large-$\beta_0$ limit of QCD where the series is exactly known to all orders in $\alpha_s$ to test the method. This was done having in mind the concrete situation of QCD, i.e., reconstructing the series solely from its first four coefficients. The method has proven to be robust and sufficiently precise to allow for a conclusion about the reliability of FOPT and CIPT, correctly reproducing the fact that FOPT is to be preferred in the large-$\beta_0$ limit. We then turned to QCD and applying the same methods reconstructed the higher orders of the series. Our main conclusions were

- The results from Padé approximants and its variants are robust. This conclusion is supported both by the tests in large-$\beta_0$ and by the fact that we are able to obtain the N3LO coefficient in QCD from the lower order ones with good precision.

- The reconstruction based on the model-independent Padé approximants favours FOPT and lends support to the renormalon models of Refs. [19, 20].

- The six-loop coefficient of the Adler function is found to be $c_{5,1} = 277 \pm 51$. This result is in line with some other estimates [19, 31], but has a smaller uncertainty.

In the remainder of this note, we will review the main results of Ref. [21] to which we refer for further details.

## 2 Overview of the theory

### 2.1 QCD in hadronic $\tau$ decays

Here, we briefly recall the main theoretical ingredients needed for the QCD analysis of hadronic $\tau$ decays. We refer to Refs. [19–21] for further details.

The main observable in hadronic $\tau$ decays is the ratio $R_\tau$ which represents the total decay width normalized to the width of $\tau \to e\bar{\nu}_e \nu_\tau$. Here, we restrict the analysis to non-strange channels which allows us to safely neglect effects due to quark masses. There are then two observables $R_{\tau,V}$ and $R_{\tau,A}$ where the decay is mediated by vector and axial-vector $\bar{u}d$ currents, respectively. They can be parametrized as

$$R_{\tau,V/A} = \frac{N_c}{2} S_{\text{EW}} |V_{ud}|^2 \left[ 1 + \delta^{(0)} + \delta_{\text{NP}} + \delta_{\text{EW}} \right], \tag{1}$$

where $S_{\text{EW}}$ and $\delta_{\text{EW}}$ are small electroweak corrections and $V_{ud}$ the CKM matrix element, $\delta_{\text{NP}}$ encloses all non-perturbative corrections both from OPE condensates and from duality-violations. The unity in between square brackets is the partonic result while $\delta^{(0)}$, which is the main object of this work, represents the perturbative QCD corrections.

The central objects in the QCD description of hadronic $\tau$ decays are the quark-current correlators defined as

$$\Pi_{V/A}^{\mu\nu}(p) \equiv i \int dx\, e^{ipx} \langle \Omega | T\{J_{V/A}^\mu(x) J_{V/A}^\nu(0)^\dagger\} | \Omega \rangle, \tag{2}$$

where $|\Omega\rangle$ is the physical vacuum and the quark currents are $J^\mu_{V/(A)}(x) = (\bar{u}\gamma^\mu(\gamma_5)d)(x)$. They admit the usual decomposition into transverse $\Pi^{(1)}_{V/A}(s)$, and longitudinal, $\Pi^{(0)}_{V/A}(s)$, parts. Because the correlators depend on conventions related to the renormalization procedure, it is advantageous to work with the Adler function, defined as the logrithmic derivative of $\Pi^{(1+0)}(s)$ as $D^{(1+0)}(s) = -s\frac{d}{ds}\left[\Pi^{(1+0)}(s)\right]$. Exploiting the analyticity of the correlators involved, the perturbative corrections are written as an integral in the complex plane with fixed $|s| = m_\tau^2$ as [19]

$$\delta^{(0)} = \frac{1}{2\pi i}\oint_{|x|=1}\frac{dx}{x}W(x)\widehat{D}^{(1+0)}_{\text{pert}}(m_\tau^2 x), \tag{3}$$

where $x = s/m_\tau^2$ and $W(x)$ is the weight function determined by kinematics. The perturbative expansion of the function $\widehat{D}$ starts at order $\alpha_s$ and can be written as

$$\widehat{D}_{\text{pert}}(s) = \sum_{n=1}^\infty a_\mu^n \sum_{k=1}^{n+1} k c_{n,k} L^{k-1}, \tag{4}$$

where $a_\mu = \alpha_s(\mu)/\pi$ and the logarithms are $L = \log(-s/\mu^2)$. The independent coefficients are the $c_{n,1}$; the others are obtained imposing renormalization group (RG) invariance in terms of the $c_{n,1}$ and $\beta$-function coefficients [19,32]. With the scale choice $\mu^2 = -s \equiv Q^2$ the logarithms can be summed to give

$$\widehat{D}(Q^2) = \sum_{n=0}^\infty r_n\alpha_s^{n+1}(Q) \equiv \sum_{n=0}^\infty c_n a_Q^{n+1}(Q) = a_Q + 1.640\,a_Q^2 + 6.371\,a_Q^3 + 49.08\,a_Q^4 + \cdots, \tag{5}$$

where $r_n = c_{n+1,1}/\pi^{n+1}$ and the numerical coefficients correspond to the choice $\mu^2 = Q^2$, $N_f = 3$, in the $\overline{\text{MS}}$ scheme.

In order to obtain the perturbative QCD corrections to $R_{\tau,V/A}$ one still needs to perform the integration shown in Eq. (3). It is then necessary to adopt a procedure to set the renormalization scale $\mu$, which appears in Eq. (4). The adoption of a running scale, $\mu^2 = Q^2$, as in Eq. (5), gives rise to the so-called Contour-Improved Perturbation Theory (CIPT), in which the $\alpha_s$ running along the contour is resummed to all orders with the $\beta$ function. In this case, the pertubative contribution $\delta^{(0)}$ can be cast as

$$\delta^{(0)}_{\text{CI}} = \sum_{n=1}^\infty c_{n,1}J_n^{\text{CI}}(m_\tau^2), \quad \text{with} \quad J_n^{\text{CI}}(m_\tau^2) = \frac{1}{2\pi i}\oint_{|x|=1}\frac{dx}{x}(1-x)^3(1+x)a^n(-m_\tau^2 x). \tag{6}$$

Another mainstream option is to adopt a fixed scale $\mu^2 = m_\tau^2$, which yields the aforementioned Fixed Order Perturbation Theory[4]. With this choice, since $\alpha_s$ is evaluated at a fixed $\mu$, the contour integrals are performed over the logarithms of Eq. (4) as

$$\delta^{(0)}_{\text{FO}} = \sum_{n=1}^\infty a_\tau^n \sum_{k=1}^n k c_{n,k}J_{k-1}^{\text{FO}}, \quad \text{with} \quad J_n^{\text{FO}} \equiv \frac{1}{2\pi i}\oint_{|x|=1}\frac{dx}{x}(1-x)^3(1+x)\ln^n(-x). \tag{7}$$

In this case, $\delta^{(0)}_{\text{FO}}$ is also an expansion in powers of the coupling whose coefficients depend on the $c_{n,1}$, on the $\beta$-function coefficients, as well as on the integrals $J_n^{\text{FO}}$. For $N_f = 3$ and in the $\overline{\text{MS}}$ scheme, the result for this expansion is

$$\delta^{(0)}_{\text{FO}} = \sum_{n=1}^\infty d_n a_Q^n = a_Q + 5.202\,a_Q^2 + 26.37\,a_Q^3 + 127.1\,a_Q^4 + (307.8 + c_{5,1})\,a_Q^5 + \cdots. \tag{8}$$

---

[4]Alternative schemes for setting the renormalization scale $\mu$ have been suggested in the literature. See, for example, Refs. [33–37].

In the last expression, the numerical result of known contributions to the first unknown coefficient are given explicitly.

Because the perturbative series is divergent, it is convenient to work with the Borel transformed series, which can have a finite radius of convergence, defined, in terms of the expansion in $\alpha_s$, as

$$B[\widehat{R}](t) \equiv \sum_{n=0}^{\infty} r_n \frac{t^n}{n!}. \tag{9}$$

The original expansion can then be understood as an asymptotic expansion to the inverse Borel transform defined as

$$\widehat{R}(\alpha) \equiv \int_0^{\infty} dt\, e^{-t/\alpha} B[\widehat{R}](t), \tag{10}$$

provided the integral exists. In our context, the series $\widehat{R}$ can represent either the reduced Adler function, $\widehat{D}$ of Eq. (5), or $\delta_{\text{FO}}^{(0)}$ given in Eq.(8). The last equation defines the sum of the asymptotic series in the Borel sense. The factorial divergence of the series in $\alpha_s$, $\widehat{D}$, leads to singularities in the $t$ variable. Two types of such singularities can be distinguished. The ultraviolet (UV) renormalons lie on the negative real axis and generate sign alternating coefficients. The infrared (IR) renormalons, on the other hand, are singularities on the positive real axis that contribute with fixed sign coefficients. IR renormalons obstruct the integration in Eq. (10) and generate an imaginary ambiguity in the inverse Borel transform. This ambiguity is expected to cancel against the power corrections in the OPE. General renormalization group (RG) arguments determine the position of the singularities in the $t$ plane. In the case of the Adler function, they appear at integer values of the variable $u \equiv \frac{\beta_1 t}{2\pi}$ (except for $u = 1$), where we denote by $\beta_1$ the leading coefficient of the QCD $\beta$ function.[5] The renormalon closest to the origin, which is the UV located at $u = -1$, dominates the behaviour of the series at large orders, which must, therefore, be sign alternating. This sign alternation is still not apparent in the coefficients of the QCD expansion that are known exactly (in the $\overline{\text{MS}}$ scheme), as seen in Eq. (5).

## 2.2 Padé approximants

Given a function $f(z)$, a rational or Padé approximant (PA) to $f(z)$, denoted $P_N^M(z)$, is the ratio of two polynomials in $z$ with order $M$ and $N$, denoted by $Q_M(z)$ and $R_N(z)$, respectively, with $R_N(0) = 1$. We consider a function $f(z)$ that assumes a series expansion around the origin in the form

$$f(z) = \sum_{n=0}^{\infty} f_n z^n. \tag{11}$$

The expansion of $P_N^M(z)$ around the origin, by construction, is the same as that of $f(z)$ for the first $M + N + 1$ coefficients. The PA $P_N^M(z)$ is therefore said to have a "contact" of order $M + N$ with the expansion of $f(z)$ around $z = 0$. Reexpanding the approximant $P_N^M(z)$ one can obtain an estimate for the the first coefficient not used as input, $f_{M+N+1}$ [25]. In this work, this type of estimate will be of special interest.

A qualitative knowledge about the analytic properties of $f(z)$ is sufficient for the successful use of Padé approximants, and with them obtain quantitative results about the function $f(z)$.

---

[5]Our definition of the QCD $\beta$-function is

$$\beta(a_\mu) \equiv -\mu \frac{da_\mu}{d\mu} = \beta_1 a_\mu^2 + \beta_2 a_\mu^3 + \beta_3 a_\mu^4 + \beta_4 a_\mu^5 + \beta_5 a_\mu^6 + \cdots$$

In fact, the PAs can be used to reconstruct the singularity structure of $f(z)$ from its expansion. Convergence theorems for this procedure exist for the cases of analytic and single-valued functions with multipoles or essential singularities [22–24]. Even for functions that exhibit branch points experience shows that the PAs can be used, in most cases, very successfully. In these specific cases, with increasing order of approximation, the poles of the approximants will tend to accumulate along the branch cut, mimicking the singularity structure of the original function [22–24].

However, the approximation of functions with cuts is more subtle. This is precisely the case of the Borel transformed Adler function in QCD. A possible strategy to circumvent the difficulties imposed by the branch points is to manipulate the series into a form more amenable to the approximation by PAs. Let us consider, for example, the case of a function $f(z) = \frac{A(z)}{(\mu-z)^\gamma} + B(z)$ which has a cut from $\mu$ to $\infty$ with exponent $\gamma$ and a reminder given by the function $B(z)$ with little structure (both $A(z)$ and $B(z)$ should be analytic at the point $z = \mu$). The method of the D-log Padé approximants [22–24] consists then in performing PAs not to $f(z)$ but to

$$F(z) = \frac{d}{dz} \log[f(z)] \sim \frac{\gamma}{\mu-z} \qquad \text{(near } z = \mu\text{)}, \tag{12}$$

which is then a meromorphic function to which the convergence theorems apply. By using appropriate Padé approximants to the function $F(z)$ one can determine in an unbiased way the pole position, $z = \mu$, as well as the residue of this pole, $-\gamma$, which, in fact, corresponds to the exponent of the branch cut of the original function $f(z)$. The values of $\mu$ and $\gamma$ are determined directly from the series coefficients. In this procedure, the approximant for $f(z)$ obtained from the PA to $F(z)$ is not necessarily a rational function. We denote by $\text{Dlog}_N^M(z)$ the Dlog-PA approximant to $f(z)$ obtained from the use of $P_N^M$ to $F(z)$ and it reads

$$\text{Dlog}_N^M(z) = f(0)e^{\int dz \frac{Q_M(z)}{R_N(z)}}, \tag{13}$$

where $P_N^M(z) = \frac{Q_M(z)}{R_N(z)}$ is, of course, the PA to $F(z)$. In practice, the approximant $\text{Dlog}_N^M(z)$ can yield a rich analytical structure, and the presence of branch cuts which are not necessarily present in the function $f(z)$ is to be expected.

# 3 Results in large-$\beta_0$

Before discussing our results in QCD, we will present results in the so-called large-$\beta_0$ limit,[6] which is a good laboratory for the strategy we present here. In a nut shell, the results in this limit are obtained considering a large number of fermion flavours, $N_f$. Then $q\bar{q}$ bubble corrections to the gluon propagator have be resummed to all orders. One then uses this dressed gluon propagator to compute the corrections with highest power of $N_f$ to all orders in $\alpha_s$ to a given QCD observable [15]. The large-$\beta_0$ limit is obtained replacing the $N_f$ dependence by $\beta_1$ in our notation, a procedure known as naive non-abelianization. Accordingly, the QCD $\beta$-function must be truncated at its first term.

The Borel transform of the reduced Adler function, shown in Eq. (9), can be written in a compact form in the large-$\beta_0$ limit as [15, 38, 39]

$$B[\widehat{D}](u) = \frac{32}{3\pi} \frac{e^{(C+5/3)u}}{(2-u)} \sum_{k=2}^{\infty} \frac{(-1)^k k}{[k^2 - (1-u)^2]^2}. \tag{14}$$

The scheme parameter $C$ is defined in such a way that the $\overline{\text{MS}}$ corresponds to $C = 0$. The result shows clearly the renormalon poles, the IR, which lie on the positive real axis, and the

---

[6]In our notation, the large-$\beta_0$ limit would be the "large-$\beta_1$" limit.

UV poles, which appear along the negative real axis. All the poles are double poles, with the exception of the IR pole closest to the origin, at $u = 2$, which is related to the gluon condensate, and is a simple pole.

It will be important to consider the Borel transform of $\delta^{(0)}$ as well which reads [19, 21]

$$B[\delta^{(0)}](u) = \frac{12}{(1-u)(3-u)(4-u)} \frac{\sin(\pi u)}{\pi u} B[\widehat{D}](u). \tag{15}$$

It is important to remark that the analytic structure of this Borel transform is now simpler than that of $B[\widehat{D}](u)$. All its UV poles are simple poles, due to the partial cancellation with the zeros of $\sin(\pi u)$. The leading IR pole of $B[\widehat{D}](u)$, at $u = 2$, is absent in $B[\delta^{(0)}](u)$ — a result first pointed out in Ref. [40, 41]. These cancellations play an important role in our analysis with PAs because the Borel transform is now much less singular. The simpler analytic structure can be much more effectively mimicked by the Pades. The leading UV pole has a residue about a factor of ten times smaller than its counterpart in the Adler function. This postpones the sign alternation of the series. PAs constructed to the expansion of Eq. (15) will benefit from these simplified analytic features of $B[\delta^{(0)}](u)$ and will lead to smaller errors yielding a better extraction of higher order coefficients [22–24].

In large-$\beta_0$, the first six coefficients of the Adler function, here denoted by $\widehat{D}_{L\beta}$, are ($N_f = 3$, $\overline{\text{MS}}$)

$$\widehat{D}_{L\beta}(a_Q) = a_Q + 1.556\, a_Q^2 + 15.71\, a_Q^3 + 24.83\, a_Q^4 + 787.8\, a_Q^5 - 1991\, a_Q^6 + \cdots, \tag{16}$$

which should be compared with their QCD counterparts shown in Eq. (5). These coefficients lead to the following $\delta^{(0)}$ in large-$\beta_0$:

$$\delta^{(0)}_{\text{FO},L\beta}(a_Q) = a_Q + 5.119\, a_Q^2 + 28.78\, a_Q^3 + 156.7\, a_Q^4 + 900.8\, a_Q^5 + 4867\, a_Q^6 + \cdots, \tag{17}$$

which can be compared with Eq. (8). As we discussed, the sign alternation is postponed and now sets in only at the 9th order due to the suppression of the UV pole in Eq. (15). In the case of the Adler function, the large-$\beta_0$ limit is a good approximation only up to $\alpha_s^2$. For $\delta^{(0)}_{\text{FO},L\beta}$, on the other hand, the approximation is good up to the last known term, i.e. $\alpha_s^4$. A reason for this better agreement in the case of the FOPT series is the fact that its coefficients depend not only on $c_{n,1}$ but also on the $\beta$-function coefficients — which are dominated by $\beta_1$ in QCD.

In Ref. [21], we have performed a careful and systematic study of the use of Padé approximants to obtain the higher-order coefficients of the series of Eqs. (16) and (17). We have verified that the procedure displays convergence and that the leading renormalon poles can be correctly reproduced. We have also discussed how renormalization scheme variations, partial Padé approximants [22–24], as well as D-log Padé approximants can be used in order to improve the quality of the approximation. Finally, we were able to design an optimal strategy to predict the higher orders based only on the first four coefficients of the series, which are the only ones available in QCD. Here we will focus on the results from this strategy.

The optimal strategy of Ref. [21] exploits the fact that the Borel transform of $\delta^{(0)}_{\text{FO}}$ displays a much simpler singularity structure. As shown in Eq. (15), this Borel transform does not have the pole at $u = 2$ and all other poles are simple poles (with the exception of the ones at $u = 3$ and $u = 4$). The UV renormalon at $u = -1$ is, in this case, more isolated from the IR singularities. It is to be expected that the use of PAs directly to this Borel transform should yield more precise results than in the case of the Adler function. We should note that a rational approximant to $\delta^{(0)}$ contains enough information for a full reconstruction of the Adler function since the independent coefficients $c_{n,1}$ can be read off from the FOPT expansion of $\delta^{(0)}$ as

$$\delta^{(0)}_{\text{FO},L\beta}(a_Q) = c_{1,1}\, a_Q + (3.563\, c_{1,1} + c_{2,1})\, a_Q^2 + (1.978\, c_{1,1} + 7.125\, c_{2,1} + c_{3,1})\, a_Q^3$$
$$+ (-45.31\, c_{1,1} + 5.934\, c_{2,1} + 10.69\, c_{3,1} + c_{4,1})\, a_Q^4 + \cdots. \tag{18}$$

We start by applying the Padé approximants to the series in $\alpha_s/\pi$, given by Eq. (17). As we have discussed, the FOPT series in large-$\beta_0$ is well behaved and, at intermediate orders, the asymptotic nature is not prominent yet. It is therefore expected that in this case the approximation of the series by PAs in $a_Q$ should lead to a good description. In Fig. 1 (lower left panel) we display an example of the results obtained (detailed numerical coefficients can be found in [21]). The agreement with the exact results is quite impressive, as seen when comparing with the upper panel of Fig. 1.

Another elegant and efficient way to obtain the higher-order coefficients is to resort to D-Log Padés constructed to the Borel transform of $\delta_{\text{FO}}^{(0)}$. This turns out to be the optimal way to improve the convergence while remaining completely model independent. The success of this strategy can be understood from the study of the function $F(u) = \frac{\text{d}}{\text{d}u} \log \big(B[\delta^{(0)}](u)\big)$, introduced in Eq. (12). The leading analytic structure of $F(u)$ is much simpler than that of the Adler function. We are left only with a leading UV simple pole at $u = -1$, an IR pole at $u = 3$ and a subleading IR pole at $u = 4$, since the pole at $u = 2$ is cancelled. It is therefore expected that the D-log Padés should perform well in the present case, since the isolated simple poles can be reproduce by the rational approximant without the need of "spending" too many coefficients.

In Fig. 1 (lower-right panel) we present results for a D-Log Padé applied to $B[\delta^{(0)}]$. The predictions for $c_{5,1}$ have a rather small relative error and the sign alternation is well reproduced by the Padés using only the first four coefficients as input. Their Borel integral provide excellent estimates for the true value of the series. However, one must note that the results from the D-Log Padés applied to $B[\delta^{(0)}]$ are less good than those of the Padés applied to series in $\alpha_s/\pi$. For example, for Dlog Padés the coefficient $c_{4,1}$ is typically wrong by a factor of about two while before it was only a few percent off. However, the approximation of the Borel transformed $\delta^{(0)}$ by D-Log Padés has the advantage that the factorial growth of the coefficients is implemented by construction and an asymptotic series is obtained, in agreement with the exact result. Furthermore, Fig. 1 shows that these small imperfections do not prevent an excellent reproduction of the exact series.

In Fig. 1 we can see that both methods allow for an excellent reproduction of the exact series up to around the 10th order. It is important to stress that the superiority of FOPT, which is well established in large-$\beta_0$, is very well reproduced in both cases even though we use as input only the first four coefficients of each series.

## 4 Results in QCD

In the large-$\beta_0$ limit, the approximants constructed to $\delta_{\text{FO}}^{(0)}$ and to $B[\delta^{(0)}]$ resulted optimal. Furthermore, for the known terms of the perturbative series for $\delta_{\text{FO}}^{(0)}$ in large-$\beta_0$ and in QCD the coefficients rather similar. This suggests that the regularity of the series is preserved in QCD, which indicates that it can be well approximated by Padé approximants constructed directly to the series in $\alpha_s/\pi$ as well. Moreover, although Eq. (15) is strictly valid only in large-$\beta_0$, since it relies on the one-loop running of the coupling, modifications to this result would arise only from higher-order beta function coefficients. It is legitimate to expect that the suppression of the leading IR singularity at $u = 2$, as well as a suppression of all the other renormalon poles with the exception of the ones at $u = 3$ and $u = 4$, should survive to a certain extent in QCD and render this Borel transform more suitable to approximation by Padé approximants.

Let us start with Padé approximants constructed to the $\alpha_s/\pi$ expansion of $\delta_{\text{FO}}^{(0)}$. We begin with a post-diction of the coefficient $c_{4,1}$ using the approximants $P_2^1(a_Q)$ and $P_1^2(a_Q)$. In Tab. 1 we display the results for six higher-order coefficients obtained from these approximants. The

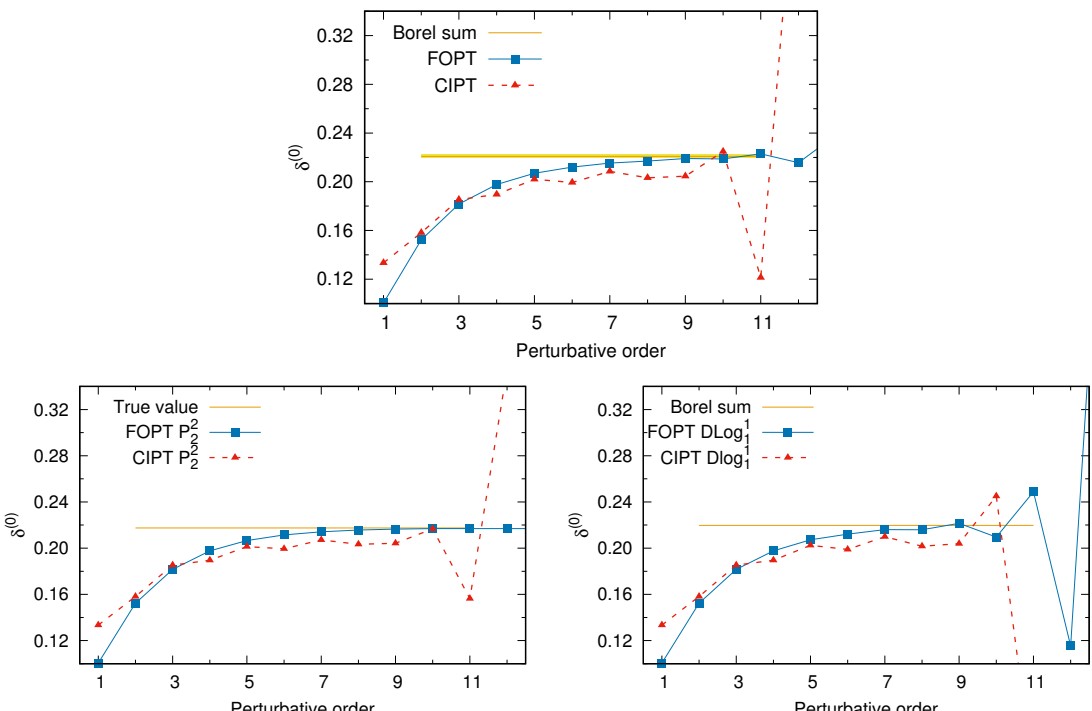

Figure 1: Perturbative expansion of $\delta^{(0)}$ in FOPT and CIPT. (Upper panel) exact large-$\beta_0$ limit, (lower-left panel) results from $P_2^2(a_Q)$, and (lower-right panel) results from $\text{Dlog}_1^1(u)$. See text for the details regarding the approximants. (The figures in the lower panels are extracted from Ref. [21].)

Table 1: QCD Adler function coefficients from PAs to the $\alpha_s$ expansion of $\delta_{\text{FO}}^{(0)}$.

|         | $c_{4,1}$ | $c_{5,1}$ | $c_{6,1}$ | $c_{7,1}$          | $c_{8,1}$          | $c_{9,1}$          | Padé sum |
|---------|-----------|-----------|-----------|--------------------|--------------------|--------------------|----------|
| $P_1^2$ | 55.62     | 276.2     | 3865      | $1.952 \times 10^4$ | $4.288 \times 10^5$ | $1.289 \times 10^6$ | 0.2080   |
| $P_2^1$ | 55.53     | 276.5     | 3855      | $1.959 \times 10^4$ | $4.272 \times 10^5$ | $1.307 \times 10^6$ | 0.2079   |
| $P_1^3$ | input     | 304.7     | 3171      | $2.442 \times 10^4$ | $3.149 \times 10^5$ | $2.633 \times 10^6$ | 0.2053   |
| $P_3^1$ | input     | 301.3     | 3189      | $2.391 \times 10^4$ | $3.193 \times 10^5$ | $2.521 \times 10^6$ | 0.2051   |

relative error in $c_{4,1}$ is only $\sim 13\%$. When put into perspective this is quite remarkable because, before the calculation of $c_{4,1}$, a forecast of this coefficient based on other methods and which included additional information (namely known terms of order $\alpha_s^4 N_f^3$ and $\alpha_s^4 N_f^2$) yielded $c_{4,1} = 27 \pm 16$ [31, 42, 43], a central value about 45% off. This gives an idea of the power of PAs.

Let us discuss now the results obtained from $P_1^3$ and $P_3^1$, also shown in Tab. 1. Now, $c_{5,1}$ is 304.7 and 301.3, respectively. We note that the results for $c_{5,1}$ and $c_{6,1}$ are strikingly stable; even those for $c_{7,1}$ and $c_{8,1}$ are remarkably similar in all approximants that we considered.

The sum of the asymptotic series performed with the PAs also leads to consistent results, as shown in the last column of Tab. 1. Our study in the large-$\beta_0$ limit strongly suggests that this stability together with the good prediction of the coefficient $c_{4,1}$ corroborate the reliability of the results. The use of D-log Padé approximants is also successful and leads to consistent results. We can, therefore, safely conclude that in QCD PAs to $\delta_{\text{FO}}^{(0)}$ are at least as reliable as in

Table 2: QCD Adler function coefficients from D-Log Padé approximants to $B[\delta^{(0)}](u)$.

|  | $c_{4,1}$ | $c_{5,1}$ | $c_{6,1}$ | $c_{7,1}$ | $c_{8,1}$ | $c_{9,1}$ | Borel sum |
|---|---|---|---|---|---|---|---|
| $\text{DLog}_0^1$ | 51.90 | 272.6 | 3530 | $1.939 \times 10^4$ | $3.816 \times 10^5$ | $1.439 \times 10^6$ | 0.2050 |
| $\text{DLog}_1^0$ | 52.08 | 273.7 | 3548 | $1.953 \times 10^4$ | $3.840 \times 10^5$ | $1.456 \times 10^6$ | 0.2052 |
| $\text{DLog}_0^2$ | input | 254.1 | 3243 | $1.725 \times 10^4$ | $3.447 \times 10^5$ | $1.187 \times 10^6$ | 0.2012 |
| $\text{DLog}_2^0$ | input | 256.4 | 3271 | $1.769 \times 10^4$ | $3.493 \times 10^5$ | $1.258 \times 10^6$ | 0.2019 |

large-$\beta_0$.

We investigate now the PAs to the Borel transform of $\delta_{\text{FO}}^{(0)}$. Although Eq. (15) is strictly valid only at one-loop, it suggests that a simplification of the analytic structure of $B[\delta^{(0)}]$ as compared to the Borel transform of the Adler function is still at work in QCD — an expectation that is supported by the results discussed below.[7] As before, the quality of the forecast of the last known coefficient, $c_{4,1}$, and the stability of the results lead us to the conclusion that the D-log Padés are again the optimal approximants to $B[\delta^{(0)}](u)$. In Tab. 2 we show higher-order coefficients extracted from D-log Padés built to $B[\delta^{(0)}](u)$.

The postdiction of $c_{4,1}$ has a relative error of only $\sim 6\%$, about a factor of two better than with Padés to the series in $\alpha_s$. The stability of the results when we include the exact value of $c_{4,1}$ as input is remarkable. Results for the unknown coefficients $c_{5,1}$ and $c_{6,1}$ are now rather stable not only among the different D-log Padés of Tab. 2, but also when we compare them with the results of Tab. 1. We should remark that all the D-log Padés of Tab. 2 predict, consistently, that the sign alternation of the series should start at the 11th order. This suggests that the leading UV singularity in QCD is less dominant than in large-$\beta_0$ which is reflected in a later sign alternation.[8] Finally, the D-log Padés can be used to predict the Borel sum of the series and the results are again very consistent (see the right-most column of Tab. 2).

The picture that emerges from the exercises performed in this section is that we have managed to obtain a very robust and reliable description of $\delta^{(0)}$ and of the Adler function at higher orders. These was achieved in a model-independent way and is consistent with the results of the large-$\beta_0$ limit.

Our final estimates for the higher-order coefficients are obtained from the approximants of Tabs. 1 and 2 including in these final values those that have only three coefficients as input parameters. This procedure allows us to take advantage of Padés that belong to different sequences and in this way we can obtain a more reliable error estimate for the final values. We do not try to favour one approximant over another and our final estimate of the coefficients and of the true value of $\delta^{(0)}$ is obtained from the average of the eight results shown in Tabs. 1 and 2.

We then attach an error equal to the maximum spread found between the results obtained from two different approximants. This error bar should give an interval where the true value of the coefficient is expected to lie, and is not an error in the statistical sense.

The application of this procedure to the six-loop coefficient, $c_{5,1}$, gives

$$c_{5,1} = 277 \pm 51. \tag{19}$$

In fact, our error estimate could even be considered as too conservative — even if smaller

---

[7]Corrections to Eq. (15) would involve terms proportional to $\beta_1/\beta_0^2$. Consequences of these terms are going to be discussed in a future work [30].

[8]This fact can be corroborate by scheme variations of the type introduced in [44]. This is discussed in detail in [21].

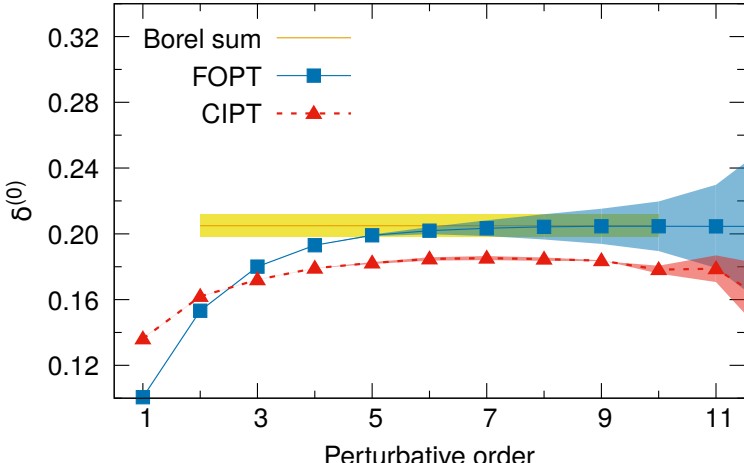

Figure 2: Final results for $\delta^{(0)}$ in QCD. The bands in the perturbative expansions stem from the uncertainty in the coefficients, while the band in the sum of the series is from the spread in the values from different PAs (see last columns of Tabs. 1 and 2). We are using $\alpha_s(m_\tau^2) = 0.316$. (Figure extracted from Ref. [21].)

than other estimates found in the literature — since it largely covers the results from all the approximants. On the basis of the available information about the Adler function, it seems unlikely that the six-loop coefficient would not lie within our bounds. As an example of other values found in the literature we have in Ref. [19] $c_{5,1} = 283 \pm 142$, from Ref. [31] one finds $c_{5,1} = 145 \pm 100$, and in Ref. [8] the value $c_{5,1} = 275$, which is remarkably close to our final central value.

With the the procedure outlined above we obtain an estimate for the true value of $\delta^{(0)}$ using the results in the last columns of Tabs. 1 and 2. With $\alpha_s(m_\tau^2) = 0.316 \pm 0.010$ [5], we find

$$\delta^{(0)} = 0.2050 \pm 0.0067 \pm 0.0130, \tag{20}$$

where the first error is our estimate from the PAs and the second error is simply due to the uncertainty in the strong coupling. This result agrees with other estimates found in the literature using other methods [19, 44–46])

Finally, we are in a position to plot the perturbative expansions of $\delta^{(0)}$ compared to the true value of the series as obtained from Eq. (20). These results are shown in Fig. 2. The bands shown in the perturbative series of Fig. 2 represent the uncertainty from the series coefficients, while the band in the Borel sum is given by the first error shown in Eq. (20). We can safely conclude that FOPT is the favored renormalization-scale setting procedure in the case of QCD. Although the CIPT series might look more stable around the fourth order, it does not approach well the central value of the sum of the series. The recommendation that FOPT is the most reliable procedure in QCD was already advocated in Refs. [19, 20] in the context of a renormalon-based model. Here it is obtained in a completely model-independent way.

## 5 Conclusion

In this work we have used the mathematical method of Padé approximants to obtain a description of the perturbative QCD series for hadronic $\tau$ decays beyond five loops. We have discussed strategies to optimize the use of the available knowledge — namely the first four

coefficients. The Borel transform of the series can be used to explain why these strategies are so efficient, as can be cross-checked from the exact results in the large-$\beta_0$ limit. The method is shown to provide accurate and reliable predictions for the higher orders and for the sum of the series. This can then be used to study the problem of renormalization-scale setting in hadronic $\tau$ decays and the result of this analysis is that fixed-order perturbation theory, FOPT, is favoured within our model-independent reconstruction of the series.

Perturbative expansions in QCD, such as the one for hadronic $\tau$ decays, are divergent series that are assumed to be asymptotic. Any conclusion about the renormalization group improvement of the series must be drawn in this context, which automatically invalidates arguments based on the "convergence" of the different series. Since a few years, there is solid renormalon-based evidence that FOPT is the best method to set the scale in $\tau$ decays [19, 20]. In this work, we have used a completely model-independent method, namely the Padé approximants, to reconstruct the higher orders [21]. Our final results are rather similar to the ones obtained from the renormalon-based methods and fully corroborate the conclusions of Refs. [19, 20]. Therefore, as of 2018, the evidence in favour of FOPT is significant and makes this procedure, most likely, the best one to be used in phenomenological studies of hadronic $\tau$ decays.

**Funding information** This work was partially supported by the São Paulo Research Foundation (FAPESP) grants 2015/20689-9 and 2016/01341-4 and by CNPq grant number 305431/2015-3. The work of P.M. is supported by the Beatriu de Pinós postdoctoral programme of the Government of Catalonia's Secretariat for Universities and Research of the Ministry of Economy and Knowledge of Spain.

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
