# Peer review of "Renormalization-group improvement in hadronic $τ$ decays in 2018"

_SciPost Physics Proceedings, doi:SciPost Phys. Proc. 1, 049 (2019)_

## Round 1 · Referee Report · Anonymous (Referee 1) · 2018-12-9

Report

This is an interesting exercise and the contribution is well written. It reports on results presented in Ref. [21], to which the authors refer for further details.

I have one major concern, and I would like the authors to address it in their contribution.

Eq. (15) is strictly valid only in the large-$\beta_0$ limit.
This means that its derivation is not forced to reproduce the two-loop universality of the QCD beta function, i.e., the fact that the first two coefficients of the QCD perturbative beta function are renormalisation-scheme independent. Let me notice that the two-loop universality has crucial physical consequences, because it determines exactly the existence of the nontrivial IR zero of QCD at large $N$ and $N_f$.

Thus my question is: how may your analysis and in particular Eq.(15) and Figure 2 change when two-loop universality of the QCD beta function is correctly accounted for in the derivation?

I would like the authors to explicitly discuss this potential issue, how and where their analysis could be affected and why eventually its numerical effects could be negligible in a certain range of $N$ and $N_f$. Most importantly, could this change the comparison between FOPT and CIPT?

Requested changes

The authors should address the issue of neglected two-loop universality, see report.

  • validity: -
  • significance: -
  • originality: -
  • clarity: -
  • formatting: -
  • grammar: -

Author:  Diogo Boito  on 2018-12-12  [id 369]

(in reply to Report 1 on 2018-12-09)
Category:
answer to question

Answer to referee's comments:

First of all, we would like to thank the referee for the careful
reading and the comments on our contribution to the proceedings.

The referee points out correctly that Eq. (15) is valid only in the
large-beta_0 limit. Therefore, effects due to the two-loop coefficient
of the beta function are absent. The modifications of this equation
when one departs from the large-beta_0 limit are two. First, the
renormalon singularities of the Adler function become branch cuts and
are no longer simple or double poles. This is discussed in detail in
our Refs. [15] and [19]. Second, the prefactor of Eq. (15) which plays
a role in the simplification of the analytic structure will change.
Actually, we are, at present, actively working on modifications to
this prefactor due to the two-loop running of alpha_s. It can be
shown, using the asymptotic formula for the running of alpha_s, that
corrections to this result are proportional to beta_2/beta_1^2.
Therefore, since the running of alpha_s is dominated by the leading
term in the beta-function, it is correct to say that in QCD Eq. (15)
is correct as a first approximation and that the main consequences of
this prefactor remain valid in QCD, i.e. the Borel transform of
delta^{0} is less singular than the Borel transform of the Adler
function.

We should point out, however, that the results in large-beta_0 are
used in our work only as a guide, a sort of laboratory for our
strategy. The results we obtain in QCD, discussed in Sec. 5, do not
rely on the large-beta_0 limit, and are obtained from the coefficients
computed at five-loops in full QCD, with no approximation. In this
sense, our final results are not affected by the limitations and
(over) simplifications that exist in the large-beta_0 limit.
Therefore, the comparison between CIPT and FOPT uses Eq. (15) only as
a guide to the general structure of the Borel transform of
delta^{(0)}, and is not affected by the details of Eq. (15).

We believe that this was not completely clear in the proceedings
because we had to shorten the discussion compared to the original
work. We have now added in Sec. 5 on p. 11 (in red) explanations on
Eq. (15) and how it is used in our final results in QCD. We believe
this new version is improved and we hope the referee will agree.

---

## Round 2 · List of Changes

Comments added in Sec. 5 to clarify the use of Eq. (15) in our analysis.

You are currently on this page

Resubmission scipost_201811_00036v2 on 12 December 2018

---

## Editorial Decision

published